# Hypoactive Visual Cortex, Prefrontal Cortex and Insula during Self-Face Recognition in Adults with First-Episode Major Depressive Disorder

**DOI:** 10.3390/biomedicines11082200

**Published:** 2023-08-04

**Authors:** Zebin Fan, Zhening Liu, Jie Yang, Jun Yang, Fuping Sun, Shixiong Tang, Guowei Wu, Shuixia Guo, Xuan Ouyang, Haojuan Tao

**Affiliations:** 1Department of Psychiatry, National Clinical Research Center for Mental Disorders, and National Center for Mental Disorders, The Second Xiangya Hospital of Central South University, Changsha 410011, China; fanzebin1995@csu.edu.cn (Z.F.);; 2Department of Radiology, The Second Xiangya Hospital of Central South University, Changsha 410011, China; 3Key Laboratory of Computing and Stochastic Mathematics (Ministry of Education), School of Mathematics and Statistics, Hunan Normal University, Changsha 410006, China; 4Key Laboratory of Applied Statistics and Data Science, College of Hunan Province, Hunan Normal University, Changsha 410006, China

**Keywords:** major depressive disorder, self-face recognition, visual cortex, task functional magnetic resonance imaging

## Abstract

Self-face recognition is a vital aspect of self-referential processing, which is closely related to affective states. However, neuroimaging research on self-face recognition in adults with major depressive disorder is lacking. This study aims to investigate the alteration of brain activation during self-face recognition in adults with first-episode major depressive disorder (FEMDD) via functional magnetic resonance imaging (fMRI); FEMDD (*n* = 59) and healthy controls (HC, *n* = 36) who performed a self-face-recognition task during the fMRI scan. The differences in brain activation signal values between the two groups were analyzed, and Pearson correlation analysis was used to evaluate the relationship between the brain activation of significant group differences and the severity of depressive symptoms and negative self-evaluation; FEMDD showed significantly decreased brain activation in the bilateral occipital cortex, bilateral fusiform gyrus, right inferior frontal gyrus, and right insula during the task compared with HC. No significant correlation was detected between brain activation with significant group differences and the severity of depression and negative self-evaluation in FEMDD or HC. The results suggest the involvement of the malfunctioning visual cortex, prefrontal cortex, and insula in the pathophysiology of self-face recognition in FEMDD, which may provide a novel therapeutic target for adults with FEMDD.

## 1. Introduction

Major depressive disorder (MDD) is a common, severe, and debilitating emotional disorder of unclear etiology. It is often considered to be one of the leading global public health problems with a high disease burden [1]. Multiple studies have shown that negative cognitive schema, especially biased self-referential cognition, play a key role in the onset, maintenance, and recurrence of depressive episodes [2]. Furthermore, growing evidence found that patients with MDD exhibit abnormal processing of self-referential processing at the experimental psychology paradigm level [3]. However, the precise mechanisms underlying biased self-referential cognition in patients with MDD remain largely unknown [3,4]. Comprehensive studies of self-referential cognition as well as brain-activity correlates of MDD may provide a new understanding of this debilitating disorder.

The self-face is the obvious and unique representation of oneself. Recognition of the self-face is not only a vital aspect of self-processing, but also a common paradigm for the systematic exploration of self-referential cognition [5,6]. A large number of studies have shown that self-face recognition has a processing advantage in speed and accuracy over familiar faces and the faces of strangers [7]. Specifically, evidence from task-state fMRI studies in recent years indicates that the participants can consistently elicit high levels of functional activities in brain areas that support face recognition and emotion processing, mainly involving the fusiform gyrus, occipital cortex, frontal gyrus, anterior and mid-cingulate, insula, putamen, and precuneus during self-face recognition [8,9]. Moreover, specific cortical regions, mainly the right inferior frontal cortex and the insula, were found to be the main regions implicated in visual self-face recognition compared with the visual processing of other persons [10,11,12]. It was proposed that self-face recognition may have a cognitive advantage in terms of a wide range of motivational, affective, and mnemonic consequences [13]. Furthermore, previous studies have shown that abnormal self-face recognition may contribute to the pathophysiology of certain mental disorders [14]. Given that self-face recognition is closely related to the individual’s level of self-awareness and mental health, we speculate that there are likely to be altered activities in the specific brain regions when individuals with MDD passively view their faces as stimuli.

Several recent studies have initially explored the neurobiology of self-face recognition among depressed adolescents via task-state fMRI and found that depressed adolescents may have neuropathological changes in the key brain regions involving the frontotemporal cortex and limbic areas [15,16,17], and that hyporesponsive self-face processing networks might be a biomarker for depression [18]. To our knowledge, neuroimaging research on self-face recognition in adults with MDD remains scarce. Considering there is significant heterogeneity between adults with MDD and depressed adolescents [19], it is uncertain whether the abnormal brain activity detected in depressed adolescents would be present in adults with MDD during the self-face stimulus.

The perception of emotional faces is a vital way to infer the individual’s psychological and physiological state and is fundamental for social interpersonal communications and adaptive behavior [20]. Recent studies indicate that neutral faces were more difficult to recognize compared with positive and negative emotional faces, and this could be due to the reduced availability of emotional information in the neutral face condition [21]. Depressed patients were particularly slow to recognize neutral faces and also recognized them less accurately than either happy or sad faces compared with healthy controls [22,23]. In addition, there is consistent evidence to suggest that depressed patients seem to perceive neutral faces as ambiguous signals of emotional neutrality and tend to attribute sadness to them [21,24]. It is worth noting that previous studies have found that the impairment in the recognition of neutral facial expressions may not only be present during depressive episodes but may also persist after remission of mood symptoms [23,24]. Therefore, we speculate that impairment of neutral facial perception might be a stable feature of depression, and exploring the potential neuropathology underpinning the deficit of neutral facial perception might be informative for understanding the neuropathological mechanism of depression. In the present study, we focus mainly on neutral face recognition in depression.

Depressed individuals often view themselves more negatively and even with disgust [25]. It is worth noting that a negative cognitive-affective schema like self-disgust is one of the common symptoms of depression and has been found to be strongly associated with the severity of depression symptoms at the experimental psychology paradigm level [26]. Anne Schienle and colleagues found that women with a higher degree of self-disgust were characterized by less grey matter volume in the bilateral insula compared to women with a low score of self-disgust [27]. The reduced insula volume may be one neural correlate of self-disgust. Previous research suggested double processing will be initiated when people view their faces, where one is the processing and integration of the face of the self at the perceptual level, and the other is the evaluation as well as an emotional response triggered by perception [9]. Therefore, we speculate that people with MDD may manifest negative emotions like disgust when they see their faces, and the degree of negative self-evaluation might link with the activation of the disgust network.

The present study was undertaken to determine whether self-neutral face recognition would trigger different brain activation between adults with first-episode MDD (FEMDD) and healthy controls (HC) via task-state fMRI. To minimize the influence of antidepressants and the course of the disease on brain function in fMRI, only adults with FEMDD and with a maximum of two weeks’ treatment with antidepressant medication were recruited in this study. Moreover, we also investigated the relationships among the group-activation differences and the severity of depressive symptoms and negative self-evaluation. We hypothesized that adults with FEMDD would show abnormal brain activation in the visual cortex, frontal cortex, and insula during the self-face-recognition tasks compared to healthy controls. We predicted that the altered brain activation would associate with the severity of depression and negative self-evaluation in participants. We believe this study will help to understand the pathological mechanism of MDD from the perspective of abnormal self-referential processing and provide a novel therapeutic target for ameliorating depression and eliminating negative self-evaluation in adults with FEMDD.

## 2. Materials and Methods

### 2.1. Participants

A total of 99 right-handed participants including 63 adults with FEMDD and 36 HC were recruited in the present study. Adults with FEMDD were recruited from the psychiatric outpatient or inpatient departments of the Second Xiangya Hospital of Central South University, while healthy controls (HC) were recruited through community advertisements. All participants were evaluated by two trained psychiatrists using the Structured Clinical Interview for the Diagnostic and Statistical Manual of Mental Disorders V Research Edition (SCID-5-RV). Participants were excluded if they met any of the following conditions: (1) Age below 18 or above 55, Han Chinese; (2) History of neurological diseases or other serious physical diseases; (3) History of electroconvulsive therapy or transcranial magnetic stimulation; (4) History of drugs, alcohol and other psychoactive substance abuse; (5) Comorbidities with other Axis I or Axis II disorders; (6) Any contraindications for MRI; (7) Alcohol or benzodiazepines consumption within 24 h before the interview and fMRI scan; (8) Pregnant or lactating, or planning to become pregnant. Participants recruited in the patient group also needed to meet the following inclusion criteria: (1) Diagnosed with a current single episode of MDD according to the Diagnostic and Statistical Manual of Mental Disorders Fifth Edition (DSM-5); (2) Total antidepressant medication duration of less than two weeks.

This study was approved by the Research Ethics Committee of Second Xiangya Hospital of Central South University (Ethical approval number: 2013 No.98). All participants provided written informed consent to participate in the study. Before consent was obtained, the capacity to provide informed consent for all potential participants was ascertained by two licensed psychiatrists. After the study procedures were explained, informed written consent was obtained from all participants. All study procedures were conducted in strict accordance with the Declaration of Helsinki.

### 2.2. Clinical Assessment

Depressive symptoms were assessed using the Hamilton Depression Rating Scale (HAMD-17). The anxiety symptoms were assessed using the Hamilton Anxiety Rating Scale (HAMA), and the general mental state of the patients was assessed by the Brief Psychiatric Rating Scale (BPRS) [28]. Negative self-evaluation was assessed with the Self-Disgust Scale (SDS) [29], which is comprised of 18 items, and the overall score ranges from 12 to 84, with higher scores indicating higher levels of self-loathing. In the present study, we adopted the Chinese version of the Self-Disgust Scale, which presents satisfactory structure and practical validity and reliability in MDD patients [30]. In addition, the mood disorder questionnaire (MDQ) [31] was used at baseline to rule out bipolar disorder as much as possible, in which any patients who answered “Yes” to 7 or more of the 13 items in question 1, “Yes” to question 2, and “Moderate” or “Serious” to question 3 would be excluded. The enrolled patients were followed up as long as possible after baseline (at least 4 months) in a face-to-face or telephone visit to the outpatient department to determine whether there was a tendency to transition to mania or hypomania. A total of 4 patients showed a tendency to transition to bipolar disorder after enrollment, and their final data were eliminated, so leaving 59 patients for subsequent analysis.

### 2.3. Experimental Task

The face-recognition task was presented using E-Prime version 2.0 (Psychology Software Tools, Sharpsburg, PA, USA, https://e-prime.software.informer.com/2.0/ (accessed on 23 December 2010)). Photographs of the participants’ faces with neutral expressions were obtained during the intake visit under standardized conditions. The neutral self-face (SNF) would be modeled by instructing the participant to recall a memory associated with neutral objects (e.g., buildings or washing machine) before a photograph with a digital camera was taken. The image of the participant’s neutral self-face was selected by visual examination and agreement between the two main experimenters. The digital photograph of each participant was edited by an image-editing program (Adobe Photoshop CS6) to remove any superfluous features such as earrings, hair, scarves, etc. (Please see the Appendix A Appendix A).

In addition, thirty-six photos of facial expressions were selected from the Chinese facial affective picture system (Preliminary Review) [32], which included 12 male photos with neutral facial expressions (other strangers’ neutral faces–male, ONF-male; please see the Appendix A Appendix A), and 12 female photos with neutral facial expressions (other strangers’ neutral faces–female, ONF-female). (Please see the Appendix A Appendix A). Additionally, 12 photos depicted disgust (other strangers’ disgusted faces, sex randomization, ODF; please see the Appendix A Appendix A), with a black background image (BCP) as control. (Please see the Appendix A Appendix A). The background detail was replaced with a flat black tone and the images were normalized in terms of spatial frequency, visual area, average brightness, and contrast. All face images were standardized (BMP format, 260 × 300 pixels, black and white). Before fMRI scanning, each participant was clearly informed that the images of facial expressions, their own and those of others, would be displayed seriatim. Participants were required to focus on watching the screen during the task.

In the scanner, participants were exposed to the images (N total = 38)–SNF, ONF, ODF, and BCP–on the screen above the eyes in seriatim display. Auditory and visual instructions that lasted for 16 s were presented before the start of the run (“The scanning is about to begin. Please keep your whole body still. After the scan starts, please focus on the picture, thank you!”).

According to the hemodynamic-response function (HRF) in fMRI, which is about 12 s and peaks after approximately 5–8 s [33], our experiment displayed each image for 9 s, with an interval of 1 s between pictures. In addition, at the end of each block, BCP was presented for 9 Sec to provide a respite from the task and establish brain-activity baselines. Subjects were instructed to press a key each time a face disappeared. (See the Appendix A Appendix A). The session ended with a blank period of 2 s, during which the scanner continued to transmit a dwindling BOLD signal. The whole task paradigm lasted 618 s and was comprised of a single run with 12 different blocks, with 5 image stimuli (SNF, ONF-male, ONF-female, ODF, and BCP) per block in a pseudorandom display. This block design followed similar procedures as those used by Semir Zeki et al. [34].

A small camera that could monitor the eyes of the participants during the scanning process confirmed that the participants did not close their eyes for more than the normal blink time during the scan, thereby verifying high-quality task completion. After the MRI scanning was over, the participants were asked to confirm whether they had been able to recognize their own photos. All the participants replied that they could clearly recognize their own photos.

### 2.4. Imaging Acquisition

A 3.0 T Siemens magnetic resonance apparatus from the Magnetic Resonance Center of the Second Xiangya Hospital of Central South University was used to obtain fMRI data. An echo-planar imaging (EPI) sequence was applied for functional scans, and the parameters were as follows: repetition time (TR) = 2000 ms, echo time (TE) = 30 ms, field of view (FOV) = 240 mm × 240 mm, flip angle (FA) = 90°; matrix = 64 × 64, axial slice = 32, and slice thickness = 4 mm. Each participant was scanned for 309-time points, with a total scanning time of 618 Sec.

### 2.5. Imaging Preprocessing

All fMRI data were preprocessed using DPABI Version 4.0_190305 (Institute of Psychology, Chinese Academy of Sciences, Beijing, China, http://rfmri.org/dpabi (accessed on 5 March 2019)) in MATLAB R2013b (8.2.0.701) (MathWorks, Natick, MA, USA, www.mathworks.com/trademarks (accessed on 13 August 2013)) [35]. Briefly, the first 8 volumes were removed to allow scanner stabilization [36]. Slice timing was used on the remaining 301 volumes to correct the phase difference caused by interlayer scanning. A realignment was then performed to reduce the impact of head movement due to physiological or autonomic factors on the image signal. Spatial normalization was further carried out and resampled to 3 × 3 × 3 mm^3^ according to the template proposed by the Montreal Neurological Institute. Finally, a Gaussian smoothing kernel with a full-width half-maximum of 8 mm was applied for spatial smoothing. Scans with a displacement of more than 3 mm along the X, Y, and Z axes and a rotation angle of more than 3.0° were not included in the final data analysis.

### 2.6. Statistical Analysis

The demographic and clinical characteristics of the FEMDD and HC were analyzed using SPSS 22.0. Age, years of education, HAMA, BPRS, HAMD, SDS and head movement parameters (mean FD Jenkinson) [37] were compared between the two groups using a one-way analysis of variance (ANOVA), and sex was compared using a chi-square test. The preprocessed fMRI data was then input into Statistic Parameter Mapping 12 software Version 6225 (University College London, UK; https://www.fil.ion.ucl.ac.uk/spm/ (accessed on 1 October 2014)) for first-level and second-level analyses. The contrast space included the whole brain area (including the cerebellum). In the first-level analysis, one-sample *t*-tests were used to compare brain activation with head-movement parameters as covariates among the following three conditions: Condition 1: SNF vs. BCP, Condition 2: SNF vs. ONF, and Condition 3: SNF vs. ODF. In the second-level analysis, a two-way analysis of covariance (ANCOVA) was used to examine the main effect of diagnosis (FEMDD and HC) and conditions respectively, and the interaction between diagnosis and conditions on brain activation adjusted by age, sex, and years of education. The DPABI viewer software was used to save the significant clusters that survived after ANCOVA analysis as the mask with correction level of the Gaussian Random Field (GRF) (voxel *p* < 0.001, cluster *p* < 0.05, two-tailed). Then, DPABI software was used to extract the Blood Oxygenation Level-Dependent (BOLD) signal value of the brain regions with significant differences in the interaction effect and simple effect of the *t*-test between the two diagnostic groups in each condition. Finally, Pearson correlation analysis was used in SPSS version 22.0 to evaluate the relationship between SDS, HAMD, and activation signal values.

### 2.7. Demographic and Clinical Characteristics

The clinical and demographic details of the samples are presented in Table 1.

Fifty-nine FEMDD (forty females, nineteen males; mean age 25.68 ± 9.61; mean education 13.88 ± 2.24 years) and thirty-six HC (twenty-two females, fourteen males; mean age 28.42 ± 8.55; mean education 16.28 ± 2.87 years) were eventually enrolled in this study. All subjects who participated in this study were Han Chinese adults.

For adults with FEMDD, the mean illness duration was 15.03 ± 19.89 months; the mean score on the Hamilton Rating Scale for Depression was 22.00 ± 4.53; the mean score on the Hamilton Rating Scale for Anxiety was 18.03 ± 6.44; the BPRS total score was 35.15 ± 7.17, and the SDS score was 58.66 ± 11.37. All patients met the DSM-5 diagnosis of MDD, had first episode of depression mood, and all were on medication for less than 2 weeks; the duration of medication was 2.53 ± 4.75 days. None of the patients had comorbidities with other disorders. For HC, the mean score on the Hamilton Rating Scale for Depression was 1.02 ± 1.90, and the SDS score was 31.35 ± 8.46. All HC participants completed the Structured Clinical Interview for Diagnostic and Statistical Manual of Mental Disorders V Research Version (SCID-5-RV) screening and were currently healthy. There was no significant difference between the two groups regarding age (*F* = 1.97, *p* = 0.16), sex (*χ*^2^ = 0.44, *p* = 0.51) and head movement parameter (*F* = 0.03, *p* = 0.87). The FEMDD’s average years of education were significantly lower than those of HC (*F* = 20.68, *p* < 0.01). In addition, the FEMDD had higher scores of HAMD (*F* = 694.90, *p* < 0.01) and SDS (*F* = 155.03, *p* < 0.01) than HC. Current and previous drug regimens (medication and duration of use) were recorded. Before the Task-fMRI scanning, 44 patients were not taking any medications, and 15 patients had been taking antidepressants and/or hypnotics (Detailed treatments for FEMDD patients can be found in Appendix A).

## 3. Results

### 3.1. Neuroimaging Results

Two-way ANCOVA showed that the interaction effect of the two diagnostic groups by three conditions was located in the left fusiform gyrus/inferior occipital gyrus (MNI [x = −36, y = −84, z = −12]; *F* = 10.06), and left calcarine/middle occipital gyrus/inferior occipital gyrus (MNI [x = −21, y = −99, z = −9]; *F* = 12.02) (Table 2, Figure 1a).

Two-way ANCOVA showed the main effects of diagnosis groups were located in the left fusiform gyrus/inferior occipital gyrus (MNI [x = −42, y = −69, z = −18]; *F* = 18.33), left superior frontal gyrus/middle frontal gyrus (MNI [x = −21, y = 54, z = 9]; *F* = 18.26), and right superior frontal gyrus/middle frontal gyrus (MNI [x = 18, y = 60, z = 27]; *F* = 16.35) (Table 2, Figure 1b).

The simple effect of the *t-*test between FEMDD vs. HC in Condition 1 (SNF vs. BCP) showed that the group difference of brain activation was located in the left occipital gyrus/fusiform gyrus (MNI [x = −42, y = −75, z = −18]; *T* = −5.14), and right occipital gyrus/fusiform gyrus (MNI [x = 45, y = −78, z = −9]; *T* = −4.57). (Table 2, Figure 1c).

The simple effect of the *t-*test between FEMDD vs. HC in Condition 2 (SNF vs. ONF) revealed that the group difference of brain activation located in the right inferior frontal gyrus/Insula (MNI [x = 45, y = 21, z = −9]; *T* = −4.45) (Table 2, Figure 1d).

The extracted brain activation values of the bilateral occipital gyrus/fusiform gyrus in condition 1(SNF vs. BCP) showed that FEMDD had significantly decreased brain activation compared with HC (Table 2, Figure 1e). Further extracted brain-activation values in the right inferior frontal gyrus/Insula in Condition 2 also showed significantly decreased brain activation in FEMDD compared with HC (Table 2, Figure 1f).

No significant difference was found for the *t*-test between FEMDD vs. HC in Condition 3 (SNF vs. ODF).

The main effects of the three conditions were located in the bilateral occipital gyrus/precuneus/fusiform gyrus (MNI [x = −18, y = −96, z = 3]; *F* = 218.70), left hippocampus/putamen (MNI [x = −21, y = −27, z = −9]; *F* = 36.92), left postcentral gyrus/precentral gyrus/frontal gyrus (MNI [x = −54, y = −21, z = 54]; *F* = 42.54), and right middle frontal gyrus/superior frontal gyrus (MNI [x = 27, y = 33, z = 36]; *F* = 19.55) (See the Appendix A).

All the above results were at the correction level of GRF: voxel *p* < 0.001, cluster *p* < 0.05, two-tailed. For detailed activation values of brain areas with interaction effect and *t-*test, please see the Appendix A Appendix A).

### 3.2. Correlations between Abnormal Brain Activation and Clinical Variables

In the FEMDD group, no significant correlation was detected between brain activation with significant group differences and the SDS scores and HAMD scores in Condition 1 (SNF vs. BCP, bilateral occipital gyrus/fusiform gyrus) or Condition 2 (SNF vs. ONF, right inferior frontal gyrus/insula). Significant correlation was also not found in HC group in Condition 1 (SNF vs. BCP, bilateral occipital gyrus/fusiform gyrus) or Condition 2 (SNF vs. ONF, right inferior frontal gyrus/insula). (See the Appendix A Appendix A.)

## 4. Discussion

The current investigation aims to identify the distinct brain activation during self-face recognition between patients with FEMDD and HC via fMRI. As predicted, decreased functional activations of the occipital cortex, fusiform cortex, prefrontal cortex, and insula were detected during self-neutral face recognition in adults with FEMDD compared with HC. Unexpectedly, no significant correlation was found between the altered brain activation and severity of depression and negative self-evaluation, neither in the FEMDD nor in the HC group. Our study provides neuroimaging evidence for abnormal activation in brain regions associated with face recognition and emotional processing in adults with FEMDD. To our knowledge, this study is the first to explore neuroimaging changes during the self-face-recognition task in adult FEMDD patients and their relationship to depression and negative self-evaluation based on task-fMRI. These findings provide new evidence that the malfunctioning visual cortex, prefrontal cortex, and insula may be involved in the neuropathology of FEMDD during self-face viewing.

The main effect of diagnosis showed dysfunctional brain activation of the occipital gyrus/fusiform gyrus/prefrontal cortex in the adults with FEMDD compared to HC, and the interaction effect indicated that the brain activation of the left occipital/fusiform area is affected by both diagnoses and conditions of face stimulation (e.g., SNF vs. BCP, SNF vs. ONF, and SNF vs. ODF). These brain areas are consistent with the regions of the face-recognition model found in previous studies [38]. Numerous studies have found that the face-recognition model was a highly developed and efficient human function, involving two main neural components called the primary visual-recognition area and the advanced information-processing area, which were also widely reported in the processing of self-face recognition [39]. Although in many previous studies, it is believed that the self-face processing model was dominated by the right side of the brain [7], several studies have also found that the importance of the left side of brain regions in self-related recognition cannot be ignored [40], especially the left fusiform gyrus [41]. The occipital lobe and fusiform gyrus are considered to be core regions of the visual face-recognition network [38]. Specifically, they are responsible for recognizing specific facial features such as eyes, nose, and mouth, perceiving and remembering different facial features and expressions, as well as distinguishing self from others [42]. In addition, the occipital lobe and fusiform gyrus have rich neural fiber connections with the prefrontal lobe, parietal lobe, temporal lobe, and insula [43,44], which are synergically involved in the regulation of self-cognitive and emotional response. Similar to other fMRI studies during visual emotive processing in MDD, our study also found abnormal activation in the prefrontal cortex [45]. Our results are also consistent with the findings that adolescents with MDD showed dysfunction of the prefrontal cortex during all conditions of the emotional self-other morph query (ESOM-Q) task compared to healthy controls [15,16]. The prefrontal cortex is responsible for advanced cognitive and emotional processes, such as controlling behavior, thinking and decision-making, emotions, and feelings of one’s own [46]. Specifically, the prefrontal region had unique neural networks for recognizing self-faces [39,47]. For instance, the dysfunction of dorsolateral-prefrontal-cortex activity was closely related to self-compassion in the self-face recognition study of adolescents with MDD [48], and the inferior frontal gyrus was found to be strongly associated with the emotional experience of recognizing or evaluating the self-face [49,50]. Moreover, the prefrontal area has been documented to be associated with self-awareness and self-control, which are both important factors involved in self-face recognition [51,52]. In contrast with previous research that reported the cortical midline structure may play an important role in self-related processing in major depressive disorder [4], the present study showed abnormal brain activation in the visual cortex and prefrontal cortex may underpin the neuropathology of MDD. However, distinct task paradigms, the heterogeneity of the participants (adolescents vs. adults), the sample size, and the effects of medicine may contribute to the inconsistency. Taken together, the findings of the main effect in the diagnostic group and interaction effect in diagnostic by face conditions provided new evidence for abnormal brain activation in the visual and prefrontal cortex at the fMRI level, which suggested that adults with FEMDD had dysfunction of both primary visual recognition and advanced information processing.

Simple effect analysis of the *t-*test between the diagnostic groups at condition 1 (SNF vs. BCP) showed significantly decreased brain activation at the bilateral occipital/fusiform areas in adults with FEMDD when they viewed neutral faces of themselves, which suggested that abnormal brain activation in the primary visual recognition area and face recognition area was present in the adults with FEMDD. It is worth noting that many studies have indicated brain areas of face recognition, usually including the occipital lobe, fusiform gyrus, and prefrontal lobe, which are recognized as core regions of the visual face-recognition network [38], no matter whether recognizing one’s own face or another’s. We speculated that the hypoactivation in occipital gyrus/fusiform gyrus in FEMDD may occur irrespective of whether it is recognition of one’s own face or the face of another person. To further clarify the issue, we compared the brain activation between FEMDD and HC during the condition of ONF vs. BCP. (For detailed information, please see Appendix A Appendix A.) Interestingly, hypoactivation in the bilateral occipital gyrus/fusiform gyrus was also detected in FEMDD. These results suggested that hypoactivation in the bilateral occipital gyrus/fusiform may be a common feature of adults with FEMDD in face recognition. In recent years, the involvement of the dysfunctional visual cortex in the pathophysiology and treatment of MDD has gradually attracted more and more attention [53]. It is worth noting that there have been studies using the visual cortex to modulate depression-like behavior in *mice* [54]. Although the exact role of abnormalities in the visual cortex in MDD remains uncertain, this underappreciated brain region might be a promising therapeutic target for adults with MDD.

Simple effect analysis of the *t-*test between diagnostic groups in condition 2 (SNF vs. ONF) showed that adults with FEMDD had decreased brain activation in the right inferior frontal gyrus/insula when they viewed self-neutral faces comparing with viewing the neutral faces of others. Previous studies have found that the inferior frontal gyrus is closely linked to the emotional experience of recognizing and evaluating self-face [50,55]. In addition, the insula was thought to be associated with the cognitive and emotional processing of self-face recognition [49]. Many studies have also highlighted the importance of the right insular region as a key node in the salience network for the coordination of internal/external information and motivation [56,57]. Specifically, there were differences in right insula activity when adolescent individuals viewed their self-faces compared to others’ faces [16]. Furthermore, the activity of the insula is correlated with self-esteem [58], self-perception, and self-awareness [59]. The abnormal brain regions identified in our study are consistent with those found in previous studies to be associated with self-face recognition. Quevedo and colleagues used an emotional self-other morph query (ESOM-Q) task, which included happy, sad, or neutral facial expressions and different morphing degrees of self and other faces to explore the neurobiology of self-face recognition in depressed adolescents and found that adolescents with MDD showed dysfunction of the prefrontal cortex and limbic areas in happy self-face versus other-face condition [15], and hypoactive mid-temporal limbic activity in happy self-face versus neutral self-face [16]. By contrast, the participants in the present study were mainly passively viewing the neutral self-face, which did not involve happy/sad self-face recognition or morphing of self/other faces. Therefore, different task paradigms and age factors may lead to inconsistent results. Our findings provide new evidence that adults with FEMDD are more impaired in self-neutral face recognition than in other-neutral face recognition compared with HC even during the task paradigm of passive viewing self-face. Furthermore, our results also suggested that the hypoactivation in the right inferior frontal gyrus/insula might be a specific neuroimaging marker for FEMDD when viewing one’s own neural face relative to viewing other-neural face.

No significant correlation was detected between brain activation with significant group differences and the severity of depression and negative self-evaluation in FEMDD or HC. The negative results may be due to the small sample size of this study, duration of disease, effects of antidepressants, educational level, or cultural background. However, it is worth noting that the insula was thought to be a key neural center for the disgust network, which plays an important role in the generation and expression of disgust [27]. Moreover, the prefrontal cortex regulates the emotion of disgust through neural connections to the insula [60]. The possible role of inferior frontal gyrus/insula in negative self-evaluation and depressive symptoms should not be ignored in future studies.

In the main effect of face conditions, this study found significant differences in brain activation in regions including the hippocampus, putamen, postcentral gyrus, and precentral gyrus when comparing self-faces with other faces or background images. These brain regions have also been found to show altered activation in self-face recognition compared with other face recognition in depressed adolescents [16,18]. The results of the main effect in face conditions supported evidence for the functional specificity of self-face recognition compared to recognition of others’ faces.

The present study has several limitations. First, this is a cross-sectional study that can only observe the current time-measurements of brain activities. Longitudinal follow-up of adult FEMDD patients may help explore the dynamic changes in brain activation during self-face recognition throughout the course of FEMDD and the potential causal relationships between brain alterations and depressive symptoms. Second, although only FEMDD patients who had been on antidepressant medication for no more than two weeks were recruited in this study, the potential effects of the medication must be considered. Third, our present study mainly focused on FEMDD and thus lacked a control group of those with recurrent MDD or other forms of depression. In future studies, we believe that comparing the brain activity related to self-face recognition among different subtypes of MDD would provide specific neural markers related to FEMDD versus depression in general. Fourth, the subjects of this study were all Han Chinese adults with different years of age and education. Although age and education were included as covariates in the analysis, the potential influence of age and educational level or even cultural backgrounds on brain activation should also be considered. In future studies, if the study participants can be categorized into different groups according to age (like young adults or non-young adults), education level or cultural backgrounds, the influence of these factors on brain activity will be minimized. Fifth, behavioral assessment or outcomes were lacking in our study, since our primary aim was to establish the neural basis of self-face recognition in FEMDD. In future studies, including behavioral assessment or outcomes may not only provide a more holistic understanding of the relationship between neuroimaging findings and real-world behavioral outcomes in FEMDD but also bridge the gap between brain function and observable behavior in depression research. Lastly, relatively small samples may limit statistical power; thus, any generalizations about the findings need to be made with caution. Further studies with larger samples are needed to improve the power of the analysis.

## 5. Conclusions

This study found that adults with FEMDD had decreased brain activation in the visual cortex, prefrontal cortex, and insula when performing self-face recognition tasks. Our study provided neuroimaging evidence for abnormal self-face information processing in MDD, which provided a new perspective for the in-depth understanding of the neuropathological mechanism of adults with FEMDD and also suggested novel therapeutic targets for MDD.

Although more and more clinical and preclinical studies have shown that MDD-related symptoms are associated with functional or structural impairments in the visual correlated processing, and antidepressants may alter the electro-physiological properties and neurotransmitters of the visual cortex, the relevant research is still in its infancy. Furthermore, it remains unclear whether the positive findings are a consequence and/or cause of MDD. Further longitudinal investigations into the neural basis of self-face recognition by combining macroscopic multimodality neuroimaging research, neurocognitive, and behavioral assessment in different subgroups of MDD may move us a step closer to the pathophysiology of this disorder. In the future, treatments such as repetitive transcranial magnetic stimulation, theta burst stimulation, and delivery of medicine to the specific brain regions can be used to directly interfere with the specific regions of the cerebral cortex to explore direct evidence for the involvement of the specific brain regions in MDD and the development of new intervention strategies.

## Figures and Tables

**Figure 1 biomedicines-11-02200-f001:**
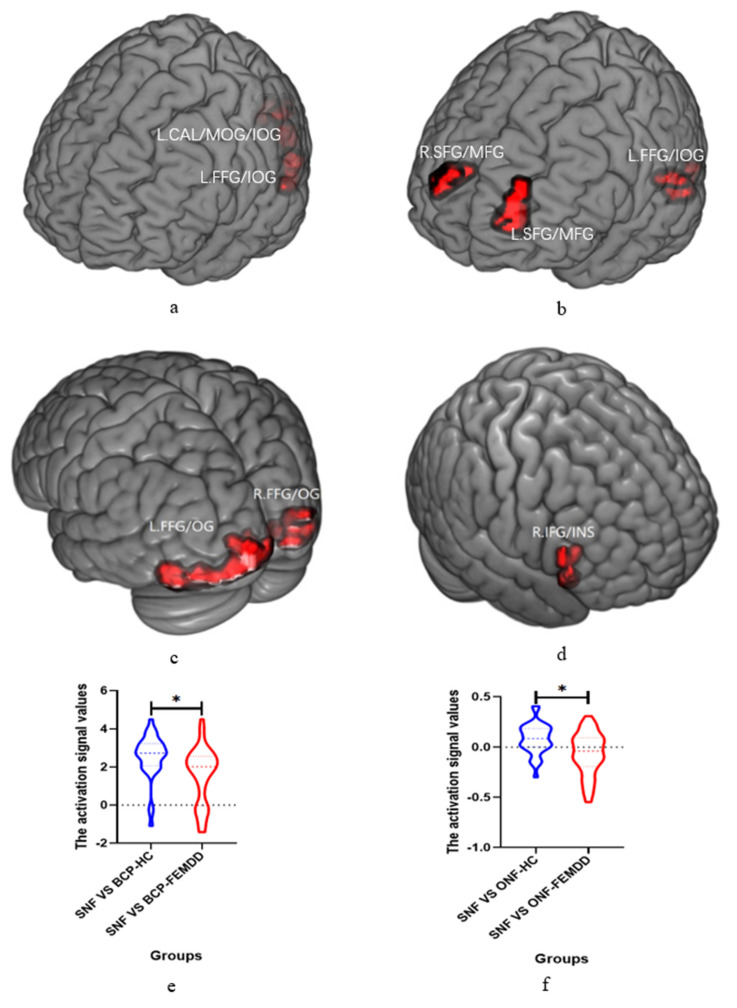
Main neuroimaging results of two−way ANCOVA with GRF correction (Voxel *p* < 0.001, Cluster *p* < 0.05, two−tailed). * represents *p*_grf_ < 0.05. (**a**) Brain areas with significant interaction effect of two groups by three conditions; (**b**) Brain areas with significant main effect of two groups; (**c**) Brain areas with significant between-group differences detected by simple effect of *t*-test in conditions 1 (SNF vs. BCP); (**d**) Brain areas with significant between-group differences detected by simple effect of *t*-test in conditions 2 (SNF vs. ONF); (**e**) The activation signal values of the brain areas with significant between-group differences detected by simple effect of *t*-test in conditions 1 (SNF vs. BCP); (**f**) The activation signal values of the brain areas with significant between-group differences detected by simple effect of *t*-test in conditions 2 (SNF vs. ONF). Abbreviation: SNF, self-neutral face; ONF, others neutral face; ODF, others disgust face; CAL, calcarine; MOG, middle occipital gyrus; IOG, inferior occipital gyrus; FFG, fusiform gyrus; IFG, inferior prefrontal gyrus; INS, insula; SFG, superior prefrontal gyrus; MFG, middle prefrontal gyrus; FEMDD, first−episode major depressive disorder; HC, healthy control; OG, occipital gyrus.

**Table 1 biomedicines-11-02200-t001:** Demographic and clinical profiles of participants in the FEMDD and HC (*n* = 95).

Variables	FEMDD (*n* = 59)	HC (*n* = 36)	Analysis	
			*F/χ* ^2^	*p* Value
Age (years)	25.68 ± 9.61	28.42 ± 8.55	1.97	0.16
Self-report sex (male/female)	19/40	14/22	0.44	0.51
Education (years)	13.88 ± 2.24	16.28 ± 2.87	20.68	<0.01
Illness course (months)	15.03 ± 19.89	-	-	-
Duration of medication (days)	2.53 ± 4.75	-	-	-
HAMD score	22.00 ± 4.53	1.02 ± 1.90	694.90	<0.01
HAMA score	18.03 ± 6.44	-	-	-
BPRS total score	35.15 ± 7.17	-	-	-
SDS score	58.66 ± 11.37	31.35 ± 8.46	155.03	<0.01
Head movement index *	0.05 ± 0.03	0.06 ± 0.02	0.03	0.87

Notes. * Head movement index: Mean FD _Jenkinson; Values are presented by mean ± standard deviation; Abbreviations: FEMDD, first-episode major depressive disorder; HC, healthy controls; HAMD, Hamilton Depression Rating Scale; HAMA, Hamilton Anxiety Rating Scale; BPRS, Brief Psychiatric Rating Scale; SDS, self-disgust scale.

**Table 2 biomedicines-11-02200-t002:** Results of two-way ANCOVA (GRF correction: Voxel *p* < 0.001, Cluster *p* < 0.05, two-tailed).

Brain Areas		Voxels	Peak MNI Coordinates (x, y, z)	*F/t* Value
Main effect of the two diagnostic groups				
Left	Fusiform gyrus/Inferior occipital gyrus	62	−42, −69, −18	18.33
Left	Superior frontal gyrus/Middle frontal gyrus	57	−21, 54, 9	18.26
Right	Superior frontal gyrus/Middle frontal gyrus	53	18, 60, 27	16.35
Main effect of three conditions				
Bilateral	Occipital gyrus/Precuneus/Fusiform gyrus	12,007	−18, −96, 3	218.70
Left	Hippocampus/Putamen	667	−21, −27, −9	36.92
Right	Middle frontal gyrus/Superior frontal gyrus	171	27, 33, 36	19.55
Left	Postcentral gyrus/Precentral gyrus/Frontal gyrus	2760	−54, −21, 54	42.54
Interaction effect: two groups by three conditions				
Left	Fusiform gyrus/Inferior occipital gyrus	41	−36, −84, −12	10.06
Left	Calcarine/ Middle occipital gyrus /Inferior occipital gyrus	100	−21, −99, −9	12.02
*t*-test between FEMDD vs. HC in condition 1				
Left	Occipital gyrus/Fusiform gyrus	436	−42, −75, −18	−5.14
Right	Occipital gyrus/Fusiform gyrus	248	45, −78, −9	−4.57
*t*-test between FEMDD vs. HC in condition 2				
Right	Inferior frontal gyrus/Insula	89	45, 21, −9	−4.45

Notes. Condition 1: SNF vs. BCP; Condition 2: SNF vs. ONF; Condition 3: SNF vs. ODF; Abbreviations: FEMDD, first-episode major depressive disorder; HC, healthy controls; SNF, self-neutral face; ONF, others-neutral face; ODF, others-disgust face.

## Data Availability

The datasets generated and analyzed during the current study are available from the corresponding author on reasonable request.

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
