# Peer review of "Hypoactive Visual Cortex, Prefrontal Cortex and Insula during Self-Face Recognition in Adults with First-Episode Major Depressive Disorder"

_biomedicines, 2023, doi:10.3390/biomedicines11082200_

Round 1

Reviewer 1 Report

The manuscript reports an evaluation of the different brain areas activated during self-face recognition between patients with depression and controls. The paper is well written. I do not have specific concerns about methodology. However, I have some comments that need to be addressed:

- have you evaluated the sample size requested for this study before the study?

- you included young adults and adults in your samples. However, there are several differences in brain's activity across ages (especially with a range from 18 to 55). You should take this aspect as a limit.

- in your sample there is a significant difference between groups as regards education. This should be included as a covariate in your analyses.

- was the participants aware that they would see their faces? Have you asked them if they recognized themselves? 

- was the cerebellum included in your analysis? (because many fMRI studies excluded it)

- please change p=0.00 into p<0.01

Reviewer 2 Report

This study investigated the alteration of brain activation during self-face recognition in adults with first-episode major depressive disorder (FEMDD) using functional magnetic resonance imaging (fMRI). The research found that individuals with FEMDD showed decreased brain activation in specific areas, including the bilateral occipital cortex, bilateral fusiform gyrus, right inferior frontal gyrus, and right insula, compared to healthy controls during the self-face recognition task. Furthermore, significant correlations were found between brain activation differences and the severity of depression and negative self-evaluation. These results suggest that malfunctioning visual cortex, prefrontal cortex, and insula may be involved in the pathophysiology of self-face recognition in FEMDD. These findings could potentially lead to the development of novel therapeutic approaches for ameliorating depression and reducing negative self-evaluation in adults with FEMDD.

The article is nicely done and provides valuable insights into self-face recognition in adults with first-episode major depressive disorder (FEMDD) using fMRI. It sheds light on the brain activation patterns associated with self-face recognition and their correlation with the severity of depressive symptoms and negative self-evaluation in individuals with FEMDD. However, it is important to note that the study focuses specifically on FEMDD and lacks a control group of individuals with recurrent major depressive disorder or those with other forms of depression. Including such additional control groups could have helped in better understanding the findings and their relevance to depression in general, beyond FEMDD. Nevertheless, the study presents promising implications for potential therapeutic targets in FEMDD, which might also have broader applications in understanding depression as a whole.

Association with Behavioral Level: While the study focused on neuroimaging outcomes, it would be valuable to discuss potential associations between the observed brain activation differences and behavioral measures relevant to depression. Addressing how the altered brain activation during self-face recognition may relate to behavioral outcomes like social functioning, emotional regulation, or cognitive biases can provide a more comprehensive understanding of the impact of FEMDD.

Reasons for Not Investigating Behavioral Level: The authors could clarify the reasons for not investigating the behavioral level in this study. Potential explanations might include limited resources, scope constraints, or the primary aim of establishing the neural basis of self-face recognition in FEMDD as a preliminary investigation. Acknowledge the importance of future studies exploring behavioral correlates to complement the current neuroimaging findings.

Potential Implications of Behavioral Investigations: In the future directions section, the authors could suggest that future research should include behavioral assessments to provide a more holistic understanding of the relationship between neuroimaging findings and real-world behavioral outcomes in FEMDD. Elaborate on the benefits of integrating neuroimaging and behavioral data to bridge the gap between brain function and observable behavior in depression research.

Demographic and clinical characteristics of the study participants belong in the "Methods" section rather than the "Results" section. In the "Methods" section, the researchers should provide a clear and detailed description of the participants' demographic information (e.g., age, gender, ethnicity, etc.) and clinical characteristics (e.g., diagnosis criteria, illness duration, severity of depression, etc.). This information is crucial to understand the sample composition and to ensure transparency and reproducibility of the study. Additionally, it helps readers to assess the generalizability of the findings to other populations or groups.

The current conclusions provide valuable insights into the altered brain activation during self-face recognition in adults with first-episode major depressive disorder (FEMDD) and its correlation with depression severity and negative self-evaluation. To enhance the comprehensiveness of the conclusions and address potential limitations, it would be beneficial for the authors to consider the following:

Expanded Discussion Section and add a \subsection on Limitations: It would be helpful to include a specific section within the "Discussion" part to elaborate on the limitations and potential implications of the findings. This section can highlight methodological constraints, such as sample size, possible confounding factors, and the cross-sectional nature of the study.

Future Directions: The authors could provide suggestions for future research directions. For instance, a longitudinal study design could help explore the dynamic changes in brain activation during self-face recognition over the course of FEMDD and potential causal relationships between brain alterations and depressive symptoms.

Comparison with Other Forms of Depression: As the study focuses on FEMDD, it would be insightful to discuss how the observed brain activation patterns compare to individuals with recurrent major depressive disorder or other forms of depression. This comparison could help discern specific neural markers related to FEMDD versus depression in general.

Clinical Implications: Expanding the conclusions to include the missing part from the abstract regarding novel therapeutic approaches for ameliorating depression and reducing negative self-evaluation in adults with FEMDD could be valuable. Discussing potential interventions or therapies that could target the identified brain regions involved in self-face recognition may enhance the clinical relevance of the study.

By incorporating these suggestions, the authors can provide a more comprehensive and insightful conclusion that addresses potential limitations, highlights future research opportunities, and explores the clinical implications of their findings.

it is ok

Reviewer 3 Report

The computation of correlation of BOLD signal with clinical scores across the whole group is invalid. These correlations were computed at brain loci that were defined on the basis of a significant difference between patients and control. At these locations, there was not a significant correlation between clincal features and brain activity in either patients or in controls. The correlation in the whole group is therefore largely due to the difference in mean value between the two groups. This difference was a consequence of the way in which the locations were defined.  The correlations tell us no more than the fact that the groups differed in brain activity during the task at these locations. As expected, symptoms were more severe in patients than controls, but we cannot conclude that the differences in brain activity were accounted for by the severity of current symptoms. 

In the discussion section, the interpretation of the observation that the data reveals decreased functional activations of the occipital cortex, fusiform cortex, prefrontal cortex, and insula were detected during self-neutral face recognition in adults with FEMDD compared with HC is very misleading. There were  differences in the viewing in brain activity at these sites when viewing image of one’s own face relative to a viewing a background without faces, but in most of these sites, there was no significant difference when viewing image of self , compared with viewing the face of another person. Thus, the reported differences at these sites cannot be attributed to process involved in recognition of one’s own face.  The evidence suggests that that most of these sites, the defect in activation occurs during viewing of faces irrespective of whether it is one’s own face or another face and also the face of another person. The only site at which there was evidence indicating deficit in brain activity when viewing one’s own face relative to viewing another face was in the right inferior frontal gyrus/insula.

The authors did observe an interaction between diagnosis by condition when all three contrasts between conditions were included in the ANOVA at multiple brain sites, but there is no way of identifying which specific contrasts between conditions accounted these interactions without examining the follow-up simple contrasts, especially the contrast of viewing one’s own face compared with viewing another face.  As mentioned in the above,  it is the right inferior frontal gyrus/insula is the only site at which there is evidence for an effect attributable to self -recognition.    

Minor issue:

line 309 The word Jenkinson is included without any explanation or definition. It might be that the authors intended to refer to the paper on Head Movement in MRI by Jenkinson et al (2002) but that paper is not  included in the reference list.

line 151: ‘satisfied structure’ should be ‘satisfactory structure’

Round 2

Reviewer 1 Report

I think the authros have addressed all my concerns

Reviewer 2 Report

It has some limitations in the design, but they are reported in the limitation section. I think the article can be accepted

English is ok

Reviewer 3 Report

The authors have dealt with my previous concerns in a satisfactory manner